# The Effect of Geometrical Non-Linearity on the Crashworthiness of Thin-Walled Conical Energy-Absorbers

**DOI:** 10.3390/ma13214857

**Published:** 2020-10-29

**Authors:** Michal Rogala, Jakub Gajewski, Miroslaw Ferdynus

**Affiliations:** Department of Machine Design and Mechatronics, Faculty of Mechanical Engineering, Lublin University of Technology, 20-618 Lublin, Poland; j.gajewski@pollub.pl (J.G.); m.ferdynus@pollub.pl (M.F.)

**Keywords:** crashworthiness, thin-walled structure, artificial neural networks

## Abstract

Crashworthiness of conical shells is known to depend on various factors. This study sets out to determine the extent to which the cross-sectional diameter contributes to their energy-absorbing properties. The object of the study was thin-walled aluminium tubes varying in upper diameter and wall thickness. The components were subjected to dynamic axial crushing kinetic energy equal to 1700 J. The numerical analysis was performed using Abaqus 6.14 software. The specific aim of the study was to determine the extent to which variable wall thickness affects the energy absorption capacity of the components under study. From the simulations, we have managed to establish a relationship between total energy absorption capacity and wall thickness. The results from the conducted analyses and the purpose-specific neural networks could provide the base for the future methodology for forecasting and optimisation of energy-absorbing systems.

## 1. Introduction

Transport safety today is an issue of critical importance, particularly given that due to technological, economic and social progress, motor vehicles have become the key everyday transport solution. The prevalence of this means of transport carries a certain danger connected with accidents, which involve heavy overloads that could result in mortal injury in people. Crush zones at the front and rear are responsible for safety in vehicles. Already in the first cars, a steel flat bar attached to the vehicle’s structural components served as a bumper protecting passengers during an accident. Nowadays, vehicles are fitted with thin-walled profiles of various cross-sections [1,2,3] attached to the front beam of a vehicle whose role is to absorb energy during an impact. The most commonly used cross-section in vehicles is the rectangle [4,5]. Because of its more stable crushing behaviour it is safer to use. Moreover, the crush initiation device is easier to make. In the case of profiles with a round cross-section, their crushing is more susceptible, so that during crushing, plastic deformation from the symmetrical (ring mode) may turn into asymmetrical, i.e., diamond, which may reduce the efficiency of the absorber [6]. An improvement of the structure of the round section was the production of conical energy absorbers. Because of the curvature of the side wall, the plastic hinges support each other in a different way, so that crushing can occur in a more predictable way. The design of the profiles ensures that they are capable of absorbing maximum energy while preventing abrupt deceleration of impact, which could otherwise result in lethal acceleration peak. 

Theoretical works analysing the course of dynamic progressive buckling began to appear [7,8]. Initial studies described thin-walled cylindrical profile degradation mechanisms [9,10]. Over the following years, scientific works focused on foam-filled thin-walled absorbers [11,12,13,14]. Due to their porous structure, these fillers increase the energy-absorption effectiveness of the absorber while maintaining the peak crushing force [15,16,17]. The energy-absorption capacity is further improved by filling the profiles with honeycomb [18,19,20,21]. The characteristics of the element cause that the peak crushing force (PCF)tends to elevate, which has an adverse effect on overloads. The crushing behaviour of thin-walled profiles is controlled using triggers [22,23,24], which may take various forms, such as notches or dents on the sidewalls or edges [25,26,27,28]. Changing the location of crush triggers affects both the shortening rate and total crush load efficiency (CLE) [29]. The improvement in energy-absorption standards of mechanical structures has focused the attention of researchers and manufacturers on passenger safety [30,31]. Excessive overloads adversely affect people in vehicles and are the leading cause of crippling injuries or even death [32,33,34]. The most susceptible structure in the human body is the brain, the damage to which has been classified in the literature as traumatic brain injury (TBI) [35,36,37]; therefore, the approach to designing energy absorbers must essentially account for maximizing energy-absorption efficiency while maintaining a safe velocity decrease rate.

Given the rapidly progressing technological advancement and the soaring data processing needs, artificial neural networks are becoming the preferred tool in processing data from numerical analyses, particularly in the use of multilayer perceptron, which is relevant for studying the relationship between the dependent variables and the output [13,38,39]. Moreover, the use of sensitivity analysis can easily determine the importance of individual dependent variables. RBF (radial basis function) and MLP (multilayer perceptron) neural networks exhibit particular efficiency in determining the relationships between INPUT-OUTPUT variables [40,41,42,43], even more so, when the studied relationships are nonlinear [44]. But for the advanced software, determining these relationships would be highly resource-consuming and in many cases impossible.

## 2. Crashworthiness Indicators

The most commonly applicable crashworthiness indicator is the force-displacement diagram (Figure 1), which presents the ratio of the crushing force and the shortening of the thin-walled profile. The behaviour of the force depicted in the graph reveals the peak crush force (PCF) and the specimen shortening over the crushing process, which provide a basic representation of the energy dissipation progression.

The key value determining the crashworthiness of a thin-walled profile is the energy absorption of the component. This quantity is represented by the field below the function graph and is obtained from Equation (1).
(1)$$EA\left(d_{x}\right)=\int_{0}^{d_{x}}F(x)dx$$
where: *d_x_*—the deformation distance, and *F*(*x*)—the crushing force expressed as a function.

The next two quantities are the peak crushing force (*PCF*), as in Figure 1, and mean crushing force (*MCF*) determined from Equation (2).
(2)$$MCF=\frac{EA(d_{x})}{d_{x}}$$

The deformation capacity of an absorber is described by the crash load efficiency (*CLE*) coefficient, defined as mean crushing force to peak crushing force ratio, derived from the following formula.
(3)$$CLE=\frac{MCF}{PCF}\times 100\%$$

The absorption capacity is further described by stroke efficiency, i.e., a quotient of the absorber length after impact (*U*) and its initial length (*L_o_*) [45].
(4)$$SE=\frac{L_{o}-U}{L_{o}}$$

The indicator that gives a comprehensive description of the absorption capacity of a thin-walled profile is total efficiency (*TE*) [11], which is obtained by multiplying *CLE* by *SE*:(5)$$TE\;\left(\%\right)=CLE\times SE$$

The final indicator of conceivably lethal overloads occurring in crushing of thin-walled absorbers is *A*, which is a multiple of the g-load.
(6)$$A=\frac{a}{g}$$
where: *a* is whole-body acceleration (or deceleration), and *g* is the gravitational acceleration.

To determine the crash severity that is inferred by A, it is necessary to employ the Macauley’s graph (Figure 2) [30] that correlates the acceleration—a multiple of gravitational acceleration—and crash pulse duration.

In the Figure above, three groups of injuries resulting from specific overloads can be discerned. In addition, there is a distinct margin between short and long crash pulse duration at 10^−1^ (s). The straight line determines from the equation TA^2.5^ = 1000 demarcates the limit acceleration, above which the human body will succumb to extensive injury.

## 3. Test Specimens

The thin-walled conical absorbers were subjected to numerical simulations. The energy-absorbing conical frustum profile under study is shown in Figure 3. The Figure presents its bottom (D1) and top (D2) diameters and sidewall thicknesses t1 and t2. The exact dimensions are given in the schematic drawing (Figure 3a) and the table (Figure 3b).

The variable thickness of the sidewall is intended to trigger profile deformation in its thinnest cross-section. In addition, the wall thickness gradation allows more energy to be absorbed at a later stage of dynamic loading [46].

The numerical analysis consisted of two stages. In the first stage, the buckling modes occur; second, the obtained modes (Figure 4) were subsequently subjected to dynamic analysis.

The boundary constraints were defined by two rigid plates placed at the bottom and top of the profile. The bottom plate was restrained by removing three translational and three rotational degrees of freedom (Figure 5a). The load applied to the reference point on the top plate was defined by these quantities: mass 70 kg, initial plate speed 7 m/s and energy of approx 1700 J.

Both plates were connected to the conical absorber by means of Tie relations. The finite element type used in numerical modelling for the non-deformable plate is R3D4 i.e., three-dimensional, 4-node quadrilateral element, whereas for the cone S4R i.e., 4-node quadrilateral shell element with reduced integration. The finite element is described by a linear shape function. The mesh size (Figure 5c) for the discretised model was 4 mm.

The elastic properties of the elastic-plastic aluminium absorber models were described by means of the Young modulus and Poisson’s ratio, and their plastic properties—by bilinear characteristics. The relevant data are presented in Table 1 below.

The material data of the aluminium alloy used for numerical analysis were obtained during a static tensile test, the results of which are shown in the stress–strain diagram below (Figure 6). The graph shows both engineering and true stresses. The true stresses are created by dividing the force by the value of the instantaneous cross-section.

In order to perfectly reproduce the boundary conditions, a stand for dynamic study was presented. The Instron CEAST 9350 High Energy System machine (Figure 7) (Norwood, MA, USA) was used as an example to show how to fix or load a specimen. In experimental conditions, the profile at its bottom has all degrees of freedom blocked by attaching it to the base of the hammer, the top part moves only in the drop direction (vertical), which is forced by the construction of the top part of the tup (it moves on the slide ways). The load has been defined as well as in the case of the true load, i.e., by defining the mass of the tup and its velocity in the range close to the performance obtained on the test stands. The boundary conditions for the numerical model corresponding to those described above are presented in Figure 5.

The impact drop tower Instron CEAST 9350 High Energy System comes with a piezoelectric sensor, installed in the lower section of the system, monitoring the impact force. In order to reproduce the experimental conditions in numerical analysis, the force was detected at a reference point located on the lower plate (Figure 5a). The capacity of the test stand is 1800 J and the maximum velocity is 24 m/s.

## 4. Numerical Analysis

Two types of numerical analysis were employed in the study: the finite element analysis, carried out with Abaqus 6.14 (Abaqus 2019, Dassault Systemes Simulia Corporation, Velizy Villacoublay, France), and modelling using artificial neural networks.

### 4.1. Finite Element Method

The FEM analysis set out to determine the effect of two variable factors on the energy-absorption properties of the models, specifically: the upper diameter (D2) and the wall thickness (t2) in the upper part of the thin-walled profile. The cross-sectional diameter of specimens changed gradually from 24 mm to 40 mm at 4-mm steps. The sidewall thicknesses were 2 mm, 3 mm and 4 mm. The numerical analyses were performed using Abaqus 6.14 software. The dynamic tests were carried out until the tup lost its entire impact velocity.

The preceding Figure presents the crushing model of a C40-28 profile that is 2-mm-thick in the upper position. From the degradation model, it may be inferred that the process was initiated in the thinnest-wall cross-section. Thus, the thickness gradation was shown to follow along the assumptions made at the design stage, acting as a spontaneous trigger.

The main source of information about the crush is the diagram (Figure 8). Below are charts for models with three gradations of sidewall thickness: 1–2 mm, 1–3 mm and 1–4 mm.

The above graph shows the characteristics for side wall gradations from 2 to 1 mm as shown in Figure 5b. The crushing behaviour of the model corresponds most closely to models with the same sidewall thickness. The maximum force value appears at the beginning and the subsequent detected force values do not exceed PCF (Figure 9). However, with such a small variation in thickness, a high value of mean crushing force can be observed, which gives a crushing force efficiency of about 60–65%.

Figure 10 and Figure 11 show how the side wall thickness gradations work. The crushing mechanism corresponds to the variable stiffness of the profile, which is shown by increasing the force with the dynamic analysis. The force peak (PCF) recorded at the beginning of the crush is a much lower value than the force in the peaks generated by the profile in further stages.

The results below were obtained from the equations detailed in Section 2. The increase in the peak crushing force (PCF) was observed to correlate with the increase in the thickness of the thin-walled profile. The correlation was, however, weaker considering the response to changes in the thickness in the upper cross-section diameter: the peak force ranged from 30 kN to 39 kN (Figure 12). The mean crushing force (MCF) exhibited a similar behaviour: its value ranged between 19 kN and 32 kN.

The contour plots below in Figure 13 provide a description of the crashworthiness performance of the studied thin-walled profiles. The CLE indicator varied from 56% to 85%. The models exhibiting the highest efficiency are represented by the red zones. The best-performing model was a design with small wall thickness and a cross-section in the range of 32–42 mm. The other effective energy-absorbing design was of larger wall thickness for the total range of diameters (D2).

The diagram on the right represents stroke efficiency (SE), whose values varied from 0.26 to 0.46. The models that have reached the most beneficial values are located in the zone marked with a dark green color. The low indicator means that the profile, despite the same initial energy, has a significant part of the profile to use to absorb additional portions of energy.

Plane diagram showing the influence of the top diameter (D2) of the pipe and the sidewall thickness (t2) on the total efficiency of numerical models. The best efficiency is achieved with models with a wall thickness of less than 3 mm (Figure 14). In addition, for diameters in the 22–32 mm range, models have better overall performance for a wider range of side wall thicknesses.

The values of the indicator in question varied: from 24 to 46 times the gravitational acceleration (Figure 15), while the crash pulse duration values were recorded in the range of 0.015 s to 0.027 s. As shown in Figure 2, for short pulse duration, the overload values *A* are not likely to result in serious injury.

### 4.2. Numerical Simulation Using theMultilayer Perceptron

The multilayer perceptron (MLP) is one of the most widely applicable artificial neural networks in scientific simulations. The purpose of its training is to define the weights (*w_ab_*). In the network layers for each input vector *x* the most accurate output *y_i_* is obtained, which will correspond to given *d_i_*. The equation defining neuron and the output signal is:$$v_{i}=f\left(\overset{N}{\underset{j=0}{\sum }}w_{ij}x_{j}\right)$$

The output signal *k* is derived from:$$y_{k}=f\left(\overset{K}{\underset{i=0}{\sum }}w_{ki}^{\prime \prime }v_{i}\right)=f\left[\sum_{i=0}^{K}w_{ki}^{\prime \prime }f\left(\overset{N}{\underset{j=0}{\sum }}w_{ij}^{\prime }x_{j}\right)\right]$$

In the presented study, the ANN was trained with back propagation. This algorithm is used in the process of training ANNs to adjust neuron weights in a multilayer network employing gradient-based optimisation methods. The performance of the ANNs was measured with the objective function known as the sum of squares (SOS), i.e., the variation between the simulated and estimated values of neural networks.

The objective function for *S* input nodes with *L* neurons in the hidden layer and *N* in the output layer is given by:$$E=\frac{1}{2}\overset{N}{\underset{k=1}{\sum }}{\left[f\left(\overset{L}{\underset{i=0}{\sum }}w_{ki}^{\prime \prime }v_{i}\right)-d_{k}\right]}^{2}=\frac{1}{2}\overset{N}{\underset{k=1}{\sum }}{\left\{f\left[\overset{L}{\underset{i=0}{\sum }}w_{ki}^{\prime \prime }\;f\left(\overset{S}{\underset{j=0}{\sum }}w_{ij}^{\prime }x_{j}\right)\right]-d_{k}\right\}}^{2}$$

The performance of the ANNs based on the available numerical data was determined to be of high quality, as confirmed by the quality indicator values oscillating at approx. 95%, and is presented in Table 2.

In addition, the testing and validation quality was also very appropriate for all neural networks. The diagrams below correlate the dependent variables with the output data (Figure 16, Figure 17 and Figure 18). The ANN simulations were carried out for MCF, CLE and TE indicators.

The results from the MLP neural network simulations provided data for the contour charts below (Figure 19), which show the energy efficiency of the absorbers. They reveal a distinctly strong relationship between the change in wall thickness and the mean crushing force (MCF). Similar behaviour is manifested by the CLE factor, which is shown to increase as the wall thickness increases.

The contour map shows that the CLE reaches its best values for small diameters and side wall thicknesses from 2.5 mm to 4 mm, and for diameters above 32 mm with the lowest thickness (Figure 20).

The TE coefficient comprehensively shows which dimensions influence the energy efficiency of the conical energy absorber most. The TE-value shows the greatest variability for side wall thickness fluctuations (Figure 21). The change in the top diameter of the absorber has less impact on TE, but models show better efficiency for diameters above 30 mm.

Table 3 shows the sensitivity analysis, which determines the influence of factors on the results of numerical analysis. The value of the indicator above 1 indicates its positive influence on the analysis. In the case under consideration, it is visible that the wall thickness gradation has a much greater impact on the obtained numerical results than the variable diameter.

## 5. Conclusions

This paper reported on the study of energy-absorption performance of conical components of variable cross-section geometry. The tested models exhibited various performance and capacity, which was observed to depend on the tested parameters t2 and D2. Peak crushing force (PCF) and mean cutting force (MCF) were correlated with the wall thickness t2: they were shown to drop in lower-wall-thickness absorbers. In addition, the relationship between the sidewall thickness (t2) and TE was noted, however, because of the behaviour of the SE coefficient under constant energy loading, these values are less important in the crush performance analysis. It was established that crush load efficiency (CLE), derived from Figure 6, is strongly dependent on the change in the diameter D2 as well as thickness t2. Moreover, t2 was found to impact the A factor, defined as a multiple of gravitational acceleration. By properly modifying D2 or t2 it is possible to reduce PCF by approximately 40% or increase CLE by approximately 60%. This behaviour of the profile with a variable thickness of the side wall allows to control the amount of absorbed energy. By changing the basic dimensions, such as the upper diameter of the cone or changing the thickness of the side walls, we can obtain an energy absorber where the crushing occurs automatically without the need to perform the triggers mechanically and with high-energy efficiency. The results obtained from the ANN simulation additionally showed a strong match between the t2 coefficient and other output data: MCF, CLE and TE. Owing to such high-quality neural networks, we can predict crush efficiency values for different D2 and t2 values. This is a tool that allowed us to verify the data obtained in a numerical way. According to the network predictions (Figure 16), the change in wall thickness t2 results in an approx. 80% higher total efficiency (TE) of the absorber. Mean crushing force (MCF) reaches the optimal performance for a diameter of 3–4 mm. In addition, the sensitivity analysis showed a much greater impact of sidewall thickness gradation on energy efficiency than a change in diameter. Future project works aim to continue the numerical verification of the data and include the experimental stage to be performed on the research stand at the Lublin University of Technology. Subsequently, MLP neural network models will be developed and employed with the purpose of designing and optimizing the energy absorption models. Finally, further studies will attempt to establish optimal locations of crush triggers in the developed energy-absorbing systems.

## Figures and Tables

**Figure 1 materials-13-04857-f001:**
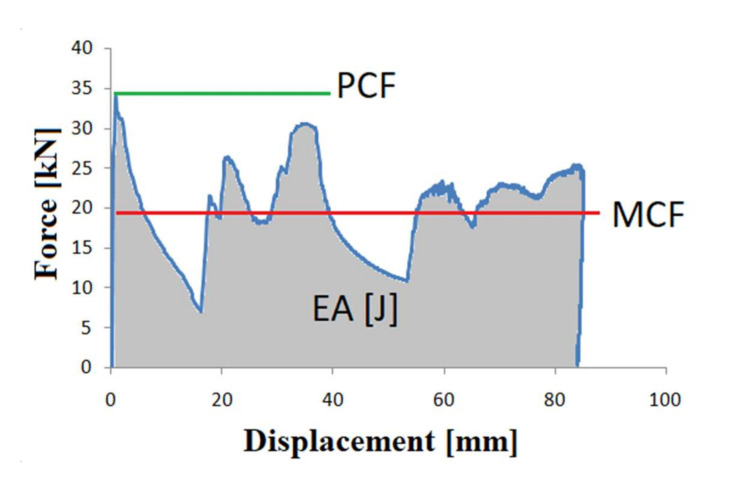
An exemplary force-displacement graph.

**Figure 2 materials-13-04857-f002:**
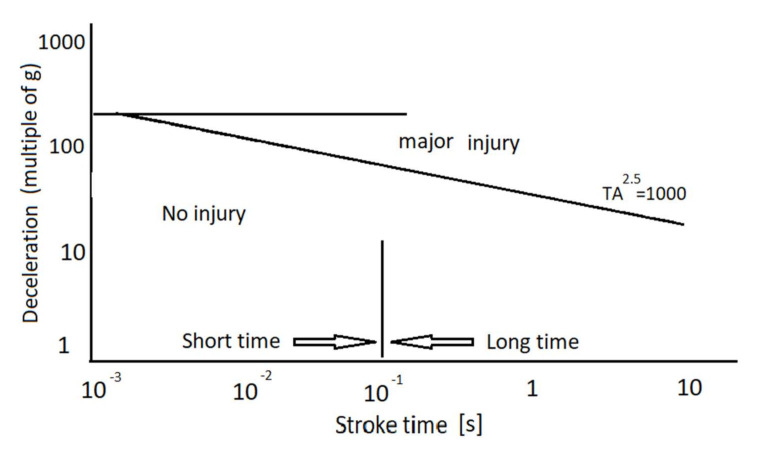
The effect of acceleration and crash pulse duration on the human body [29,30].

**Figure 3 materials-13-04857-f003:**
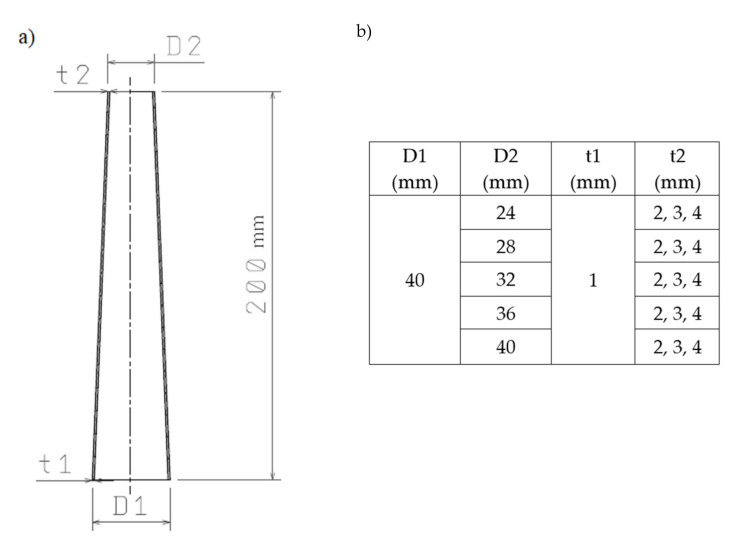
(**a**) The schematic drawing of the profile; (**b**) Variable cone parameters.

**Figure 4 materials-13-04857-f004:**
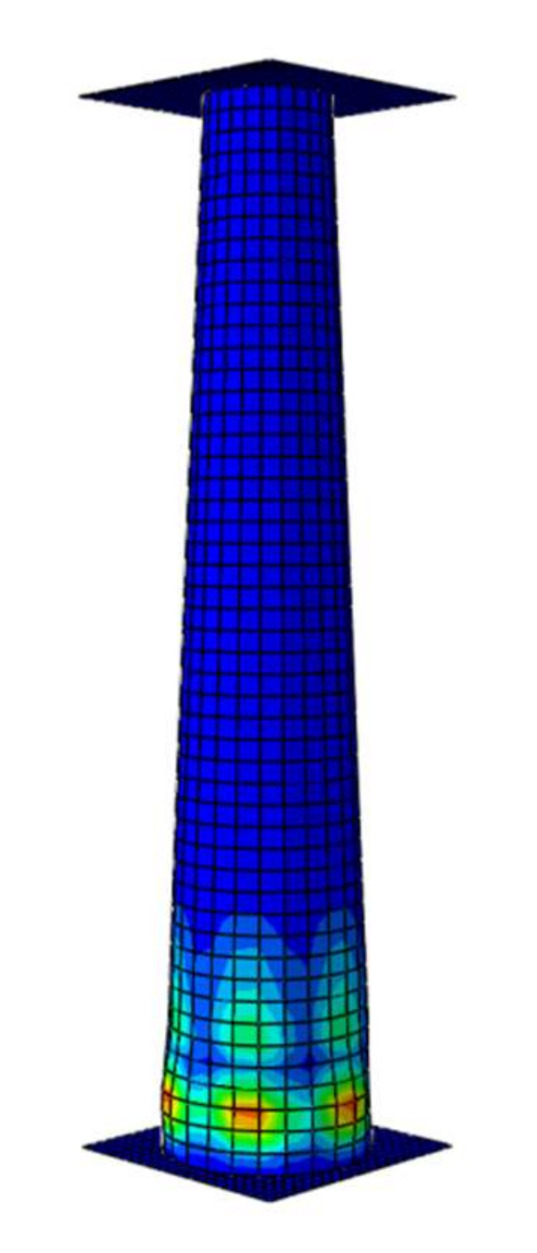
The form of buckling of the model with a thickness of 1–2 mm.

**Figure 5 materials-13-04857-f005:**
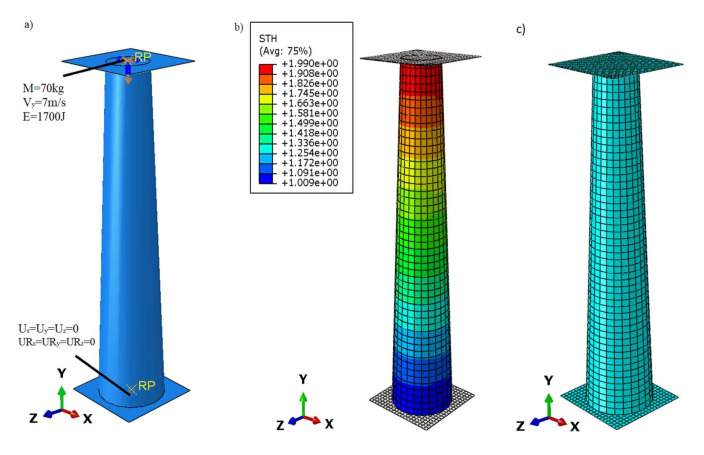
(**a**) Boundary constraints defined in finite element analysis; (**b**) the variable wall thickness of the tested profile; (**c**) discrete numerical model.

**Figure 6 materials-13-04857-f006:**
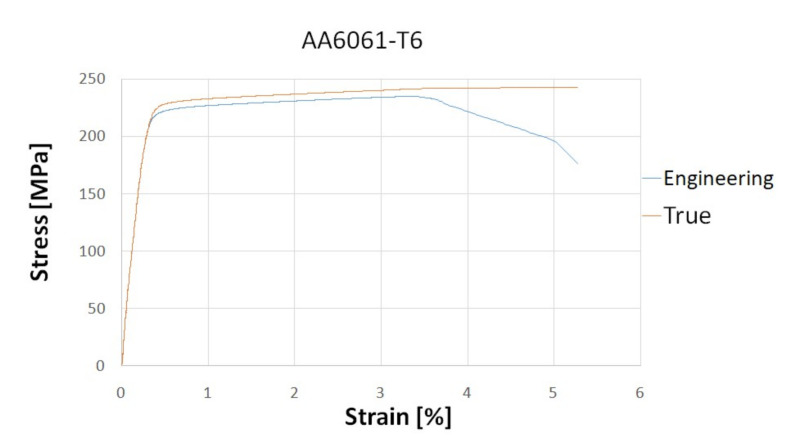
Stress–strain curve for aluminium alloy.

**Figure 7 materials-13-04857-f007:**
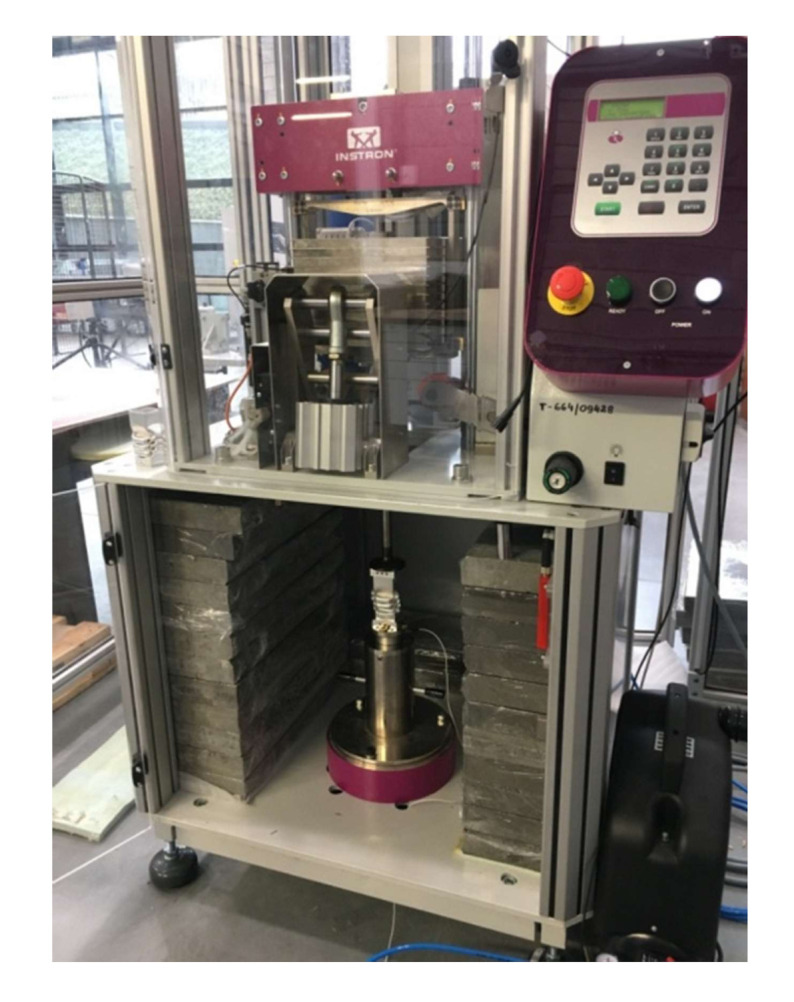
The impact drop tower for dynamic loading experiments.

**Figure 8 materials-13-04857-f008:**
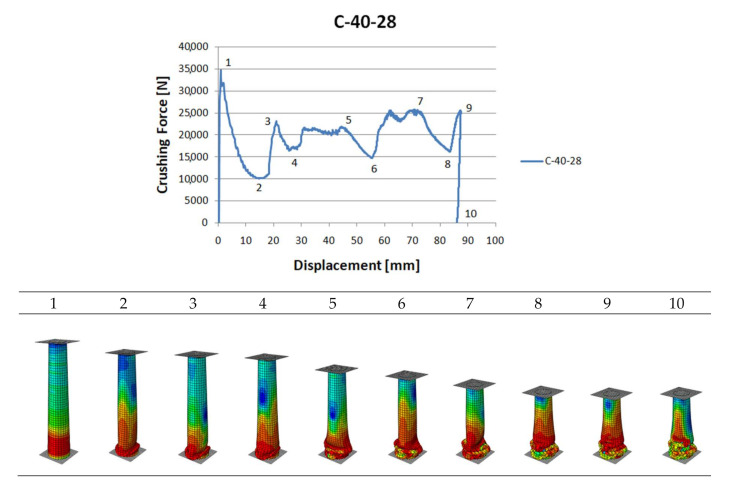
Crushing force—shortening characteristics and different crushing modes.

**Figure 9 materials-13-04857-f009:**
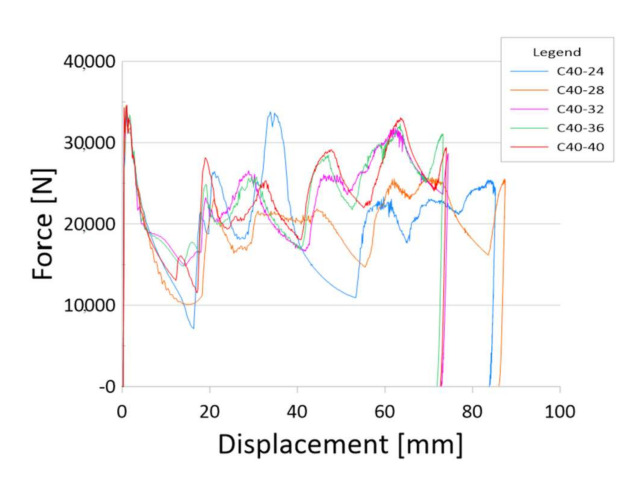
The results for models with a sidewall thickness of 1–2 mm.

**Figure 10 materials-13-04857-f010:**
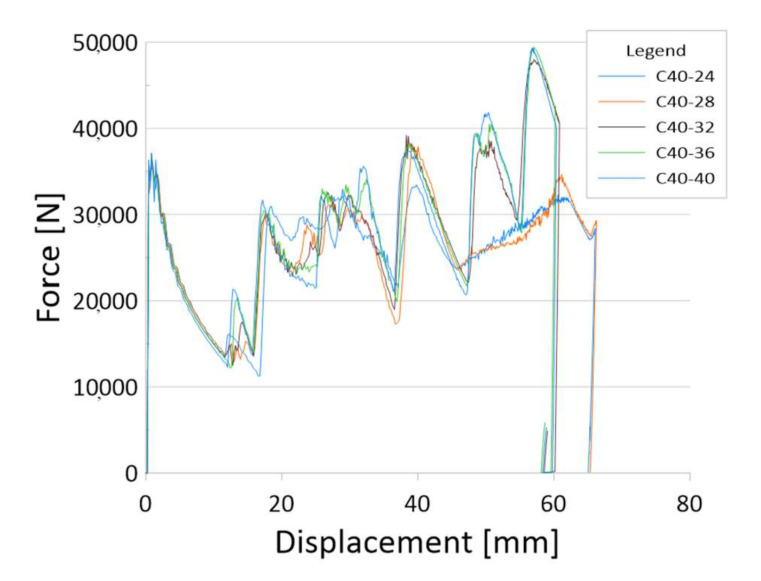
The results for models with a sidewall thickness of 1–3 mm.

**Figure 11 materials-13-04857-f011:**
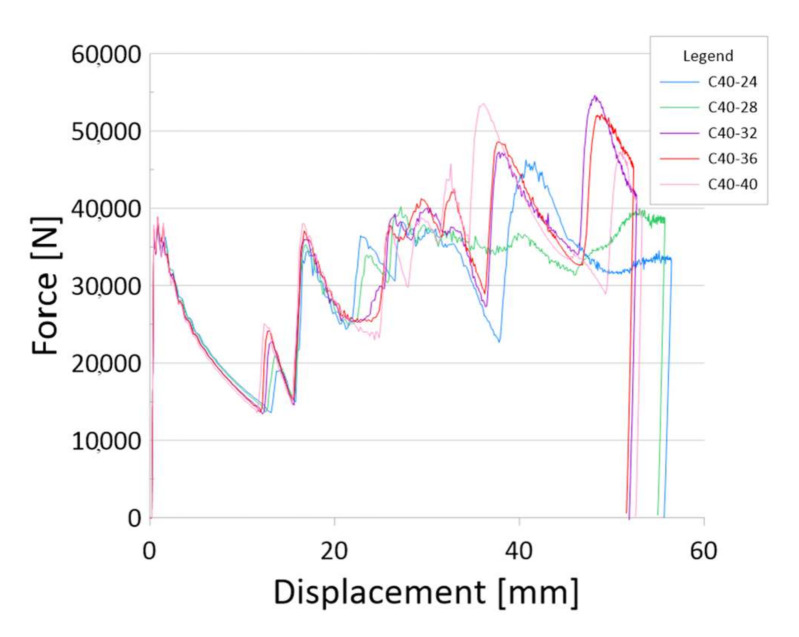
The results for models with a sidewall thickness of 1–4 mm.

**Figure 12 materials-13-04857-f012:**
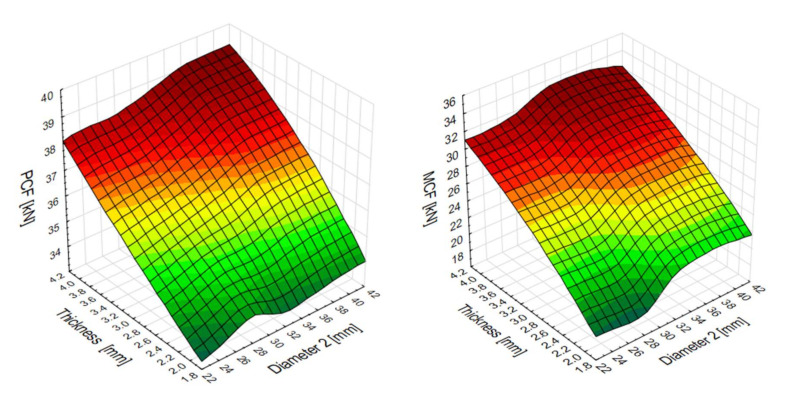
Peak crushing force (PCF) and mean crushing force (MCF) surface plots.

**Figure 13 materials-13-04857-f013:**
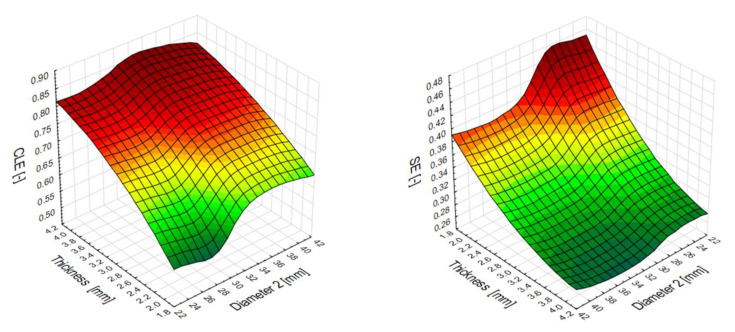
Crush load efficiency (CLE) and stroke efficiency (SE) indicators from the numerical models.

**Figure 14 materials-13-04857-f014:**
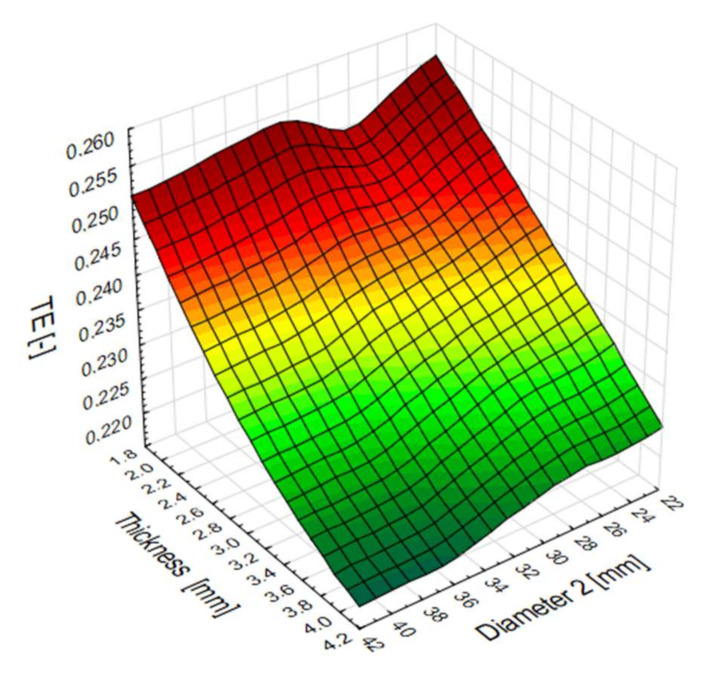
Diagram of the total efficiency (TE) indicator for a specific data obtained in numerical analysis.

**Figure 15 materials-13-04857-f015:**
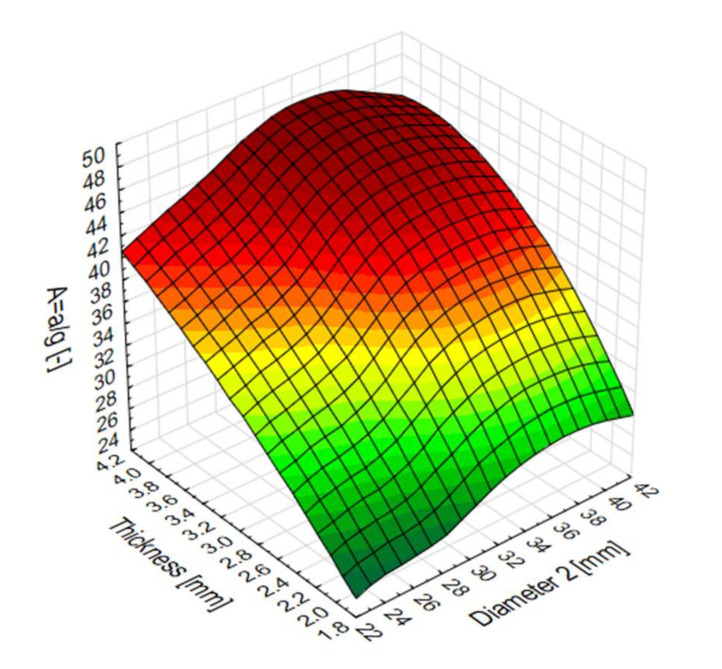
The g-load multiple (A) surface plot for numerical models.

**Figure 16 materials-13-04857-f016:**
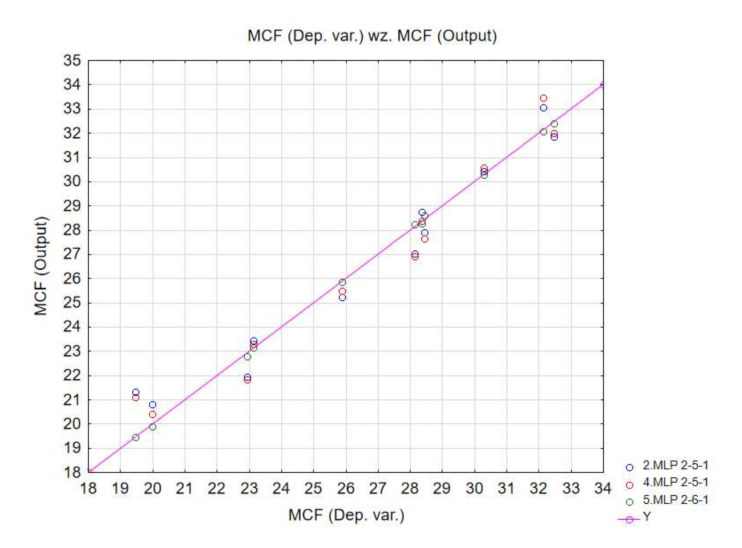
MCF neural network training quality.

**Figure 17 materials-13-04857-f017:**
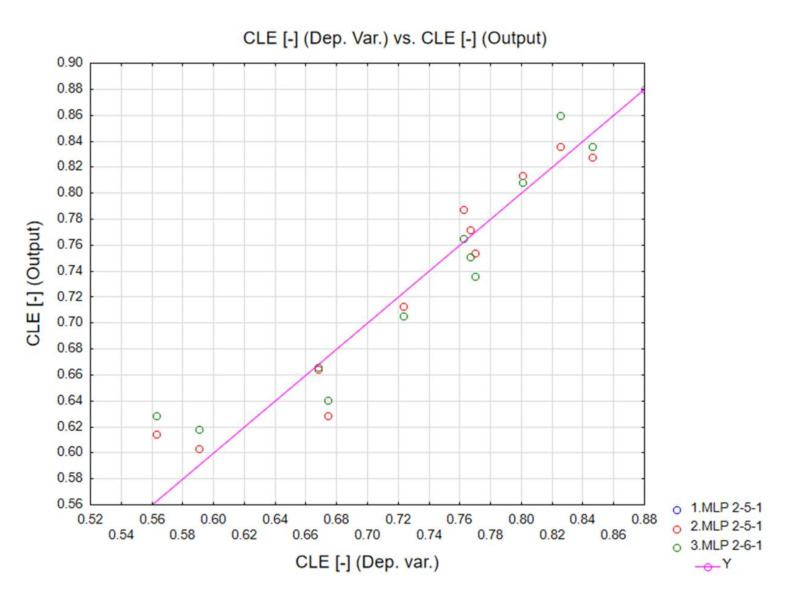
CLE neural network training quality.

**Figure 18 materials-13-04857-f018:**
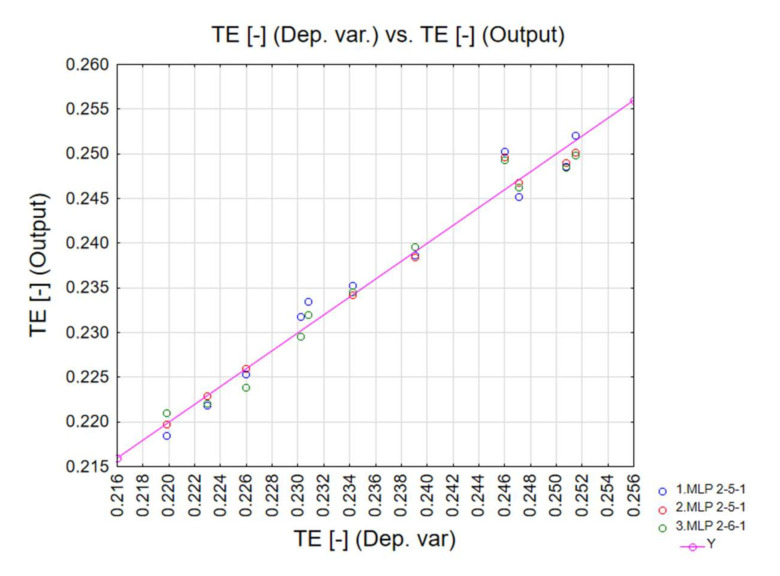
TE neural network training quality.

**Figure 19 materials-13-04857-f019:**
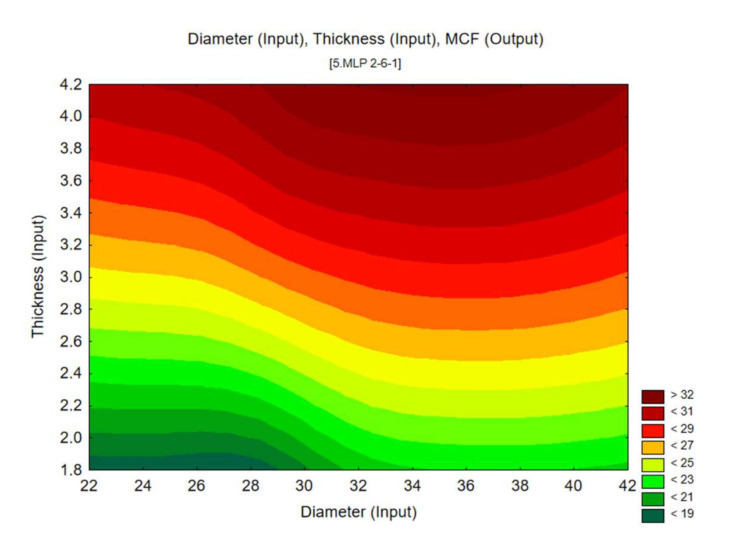
Mean crushing force (MCF) predicted by MLP 2-6-1 artificial neural network.

**Figure 20 materials-13-04857-f020:**
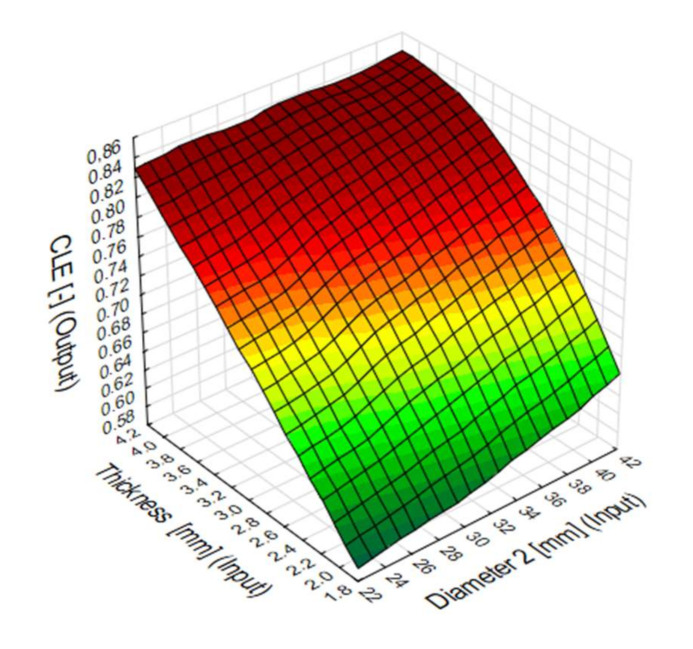
Crush load efficiency (CLE) predicted by MLP 2-4-1 artificial neural network.

**Figure 21 materials-13-04857-f021:**
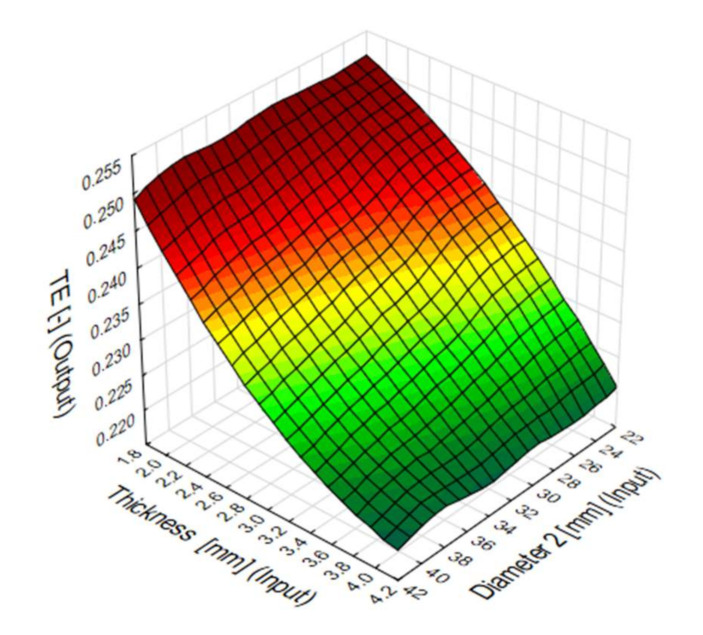
Total efficiency (TE) predicted by MLP 2-7-1 artificial neural network.

**Table 1 materials-13-04857-t001:** Aluminium model properties.

AA6061-T6 Aluminium Material Properties	
Density (kg/m^3^)	2700
Young Modulus (MPa)	70,000
Poisson ratio v (-)	0.33
Yield point Re (MPa)	200
Tensile Strength Rm (MPa)	279.98
Elongation E (%)	5.98

**Table 2 materials-13-04857-t002:** Quality of neural networks presented in the paper.

Network	Quality (Training)	Quality (Testing)	Quality (Validating)	Error (Training)	Learning Algorithm	Error	Activation (Hidden)	Activation (Output)
MLP 2-5-1	0.96102	0.95473	0.96421	0.000619	BFGS 8	SOS	Linear	Tahn
MLP 2-5-1	0.98577	0.98312	0.97432	0.000229	BFGS 38	SOS	Exponential	Logistic
MLP 2-6-1	0.99392	0.97501	0.97261	0.000970	BFGS 6	SOS	Gaussa	Exponential

**Table 3 materials-13-04857-t003:** Sensitivity analysis carried out using neural networks.

Networks	Thickness	Diameter
MLP 2-5-1	43.0670	4.79858
MLP 2-5-1	162.6045	26.37700
MLP 2-6-1	25.9865	2.61329
Average	77.2193	11.26296

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
