# Peer review of "The Effect of Geometrical Non-Linearity on the Crashworthiness of Thin-Walled Conical Energy-Absorbers"

_materials, 2020, doi:10.3390/ma13214857_

Round 1

Reviewer 1 Report

Please find attached the manuscript with included comments and suggestions from my review

Reviewer 2 Report

The current manuscript proposes a study on the effects of cross-sectional parameters on the structural properties of thin-walled conical tube, whose crashworthiness was numerically investigated under axial crushing loading. The results showed that the upper diameter and wall thickness have significant impacts on the structural performances. However, the novelty of the manuscript is not obvious. The authors only used the commercial FE software to calculate the MCF, PCF, CLE, TE, and SE of the conical tubes under axial crushing loading and employed well-known surrogate model such as ANN. The accuracy of FE model is not validated, which makes the following parametric study questionable. Moreover, the prediction accuracies of the surrogate models are not provided. Specific comments are as follows:

Q1. May I suggest the authors to do some comprehensive literature review on energy absorption structures with variable cross-section geometry and conical shape, which are missing from the Introduction part? The academic rigor of the research background should also be improved. In addition, the Introduction part is suggested to be stated in a more logical way.

Q2. Please deliver the contents clearly and concisely.

Q3. Full names of the abbreviations used in the manuscript should be given when they first appear.

Reviewer 3 Report

In the paper THE EFFECT OF GEOMETRICAL NON-LINEARITY ON THE CRASHWORTHINESS OF THIN-WALLED CONICAL ENERGY-ABSORBERS, the extent to which the variable wall thickness affects the energy absorption capacity of the studied components was analyzed.
Two types of numerical analyzes were performed in this paper, namely, the first one: the finite element analysis, carried out with Abaqus 6.14, and the second one modelling using artificial neural networks.

For numerical analysis authors used as model aluminium alloy A6061 type (the elastic specific properties was used as material data). After the static tensile test, according to with Stress-Strain curve for aluminium alloy, the experimental results are in concordance with material data used for comparison.
The data obtained from the dynamic tests were applied to produce the load-shortening diagram, which enabled generating crashworthiness indicators and the profiles were verified with a dynamic discharge machine.
The paper emphasizes that the crushing behaviour of the model corresponds most closely to that of models with the same thickness of the sidewall. The value of the maximum force appears at the beginning, and the values of the force detected later do not exceed PCF. However, with such a small variation in thickness, we can observe a characteristically high course, which results in an efficiency of about 60-65%.

The numerical simulation in the case of the artificial network was made using the Multilayer Perceptron (MLP). The results from the MLP neural network simulations provided data for the contour charts below, which show the energy efficiency of the absorbers. They reveal a distinctly strong relationship between the change in wall thickness and the mean crushing force (MCF). Similar behaviour is manifested by the TE factor, which is shown to decrease as the wall thickness increases.
In conclusions, all the computational studies were validated by experiments.

The paper, as a whole, is of interest, especially for researchers interested in computational materials research (application of modern computational methods with experimental techniques). Also, the information's obtained from this study is of real use on the future measures and design of vehicles to ensure more protection and safety of drivers and passengers in traffic, in case of accidents.

Author Response

Thank you very much for your rightful review. I am glad that my work has been so appreciated by you. The manuscript in the corrected form was uploaded to the system. Thank you once again for a thorough analysis of my work.

Round 2

Reviewer 2 Report

The paper can be accepted for publication in its present form.

Author Response

Thank you for your valuable review and for reviewing the revised version of the manuscript. Thank you very much for your positive evaluation of the presented content and acceptance of the work in its current form.